# Thapsigargin Is a Broad-Spectrum Inhibitor of Major Human Respiratory Viruses: Coronavirus, Respiratory Syncytial Virus and Influenza A Virus

**DOI:** 10.3390/v13020234

**Published:** 2021-02-03

**Authors:** Sarah Al-Beltagi, Cristian Alexandru Preda, Leah V. Goulding, Joe James, Juan Pu, Paul Skinner, Zhimin Jiang, Belinda Lei Wang, Jiayun Yang, Ashley C. Banyard, Kenneth H. Mellits, Pavel Gershkovich, Christopher J. Hayes, Jonathan Nguyen-Van-Tam, Ian H. Brown, Jinhua Liu, Kin-Chow Chang

**Affiliations:** 1School of Veterinary Medicine and Science, University of Nottingham, Sutton Bonington, Nottingham LE12 5RD, UK; sarah.al-beltagi@nottingham.ac.uk (S.A.-B.); cristian.preda@nottingham.ac.uk (C.A.P.); belinda.wang@nottingham.ac.uk (B.L.W.); jiayun.yang@nottingham.ac.uk (J.Y.); 2The Pirbright Institute, Ash Road, Pirbright, Woking GU24 0NF, UK; leah.goulding@pirbright.ac.uk; 3Animal and Plant Health Agency (APHA), Weybridge, Woodham Lane, Addlestone KT15 3NB, UK; joe.james@apha.gov.uk (J.J.); paul.skinner@apha.gov.uk (P.S.); ashley.banyard@apha.gov.uk (A.C.B.); ian.brown@apha.gov.uk (I.H.B.); 4Key Laboratory of Animal Epidemiology, Ministry of Agriculture, College of Veterinary Medicine, China Agricultural University, 2 Yuanmingyuan West Road, Beijing 100193, China; 08024h@cau.edu.cn (J.P.); jiang_zm@cau.edu.cn (Z.J.); ljh@cau.edu.cn (J.L.); 5School of Biosciences, University of Nottingham, Sutton Bonington, Nottingham LE12 5RD, UK; ken.mellits@nottingham.ac.uk; 6School of Pharmacy, University of Nottingham, University Park, Nottingham NG7 2RD, UK; pavel.gershkovich@nottingham.ac.uk; 7School of Chemistry, University of Nottingham, University Park, Nottingham NG7 2RD, UK; chris.hayes@nottingham.ac.uk; 8School of Medicine, University of Nottingham, University Park, Nottingham NG7 2RD, UK; jvt@nottingham.ac.uk

**Keywords:** thapsigargin, inhibitor, antiviral, SARS-CoV-2, coronavirus OC43, respiratory syncytial virus, influenza virus, broad-spectrum, innate immunity, mouse, remdesivir, ribavirin, oseltamivir

## Abstract

The long-term control strategy of SARS-CoV-2 and other major respiratory viruses needs to include antivirals to treat acute infections, in addition to the judicious use of effective vaccines. Whilst COVID-19 vaccines are being rolled out for mass vaccination, the modest number of antivirals in use or development for any disease bears testament to the challenges of antiviral development. We recently showed that non-cytotoxic levels of thapsigargin (TG), an inhibitor of the sarcoplasmic/endoplasmic reticulum (ER) Ca^2+^ ATPase pump, induces a potent host innate immune antiviral response that blocks influenza A virus replication. Here we show that TG is also highly effective in blocking the replication of respiratory syncytial virus (RSV), common cold coronavirus OC43, SARS-CoV-2 and influenza A virus in immortalized or primary human cells. TG’s antiviral performance was significantly better than remdesivir and ribavirin in their respective inhibition of OC43 and RSV. Notably, TG was just as inhibitory to coronaviruses (OC43 and SARS-CoV-2) and influenza viruses (USSR H1N1 and pdm 2009 H1N1) in separate infections as in co-infections. Post-infection oral gavage of acid-stable TG protected mice against a lethal influenza virus challenge. Together with its ability to inhibit the different viruses before or during active infection, and with an antiviral duration of at least 48 h post-TG exposure, we propose that TG (or its derivatives) is a promising broad-spectrum inhibitor against SARS-CoV-2, OC43, RSV and influenza virus.

## 1. Introduction

In the midst of COVID-19 pandemic, the need for effective vaccines for mass vaccination has never been greater to control and prevent rampant morbidity, mortality and surging clinical cases that can overwhelm hospitals. COVID-19 is caused by a novel coronavirus, designated severe acute respiratory syndrome coronavirus 2 (SARS-CoV-2), an enveloped positive single-strand RNA virus [1]. Although COVID-19 vaccines are now in deployment, eradication is unrealistic and there is an unmet parallel need for effective SARS-CoV-2 antivirals [2,3] to inhibit active virus replication in patients to reverse virus progression. Given that not all vaccines will necessarily prevent virus shedding in subsequent infection, antivirals could also serve to reduce the overall virus levels circulating in a population, thus reducing its spread. The modest number of antivirals in use or in development for any infectious disease bears testament to the practical challenges of antiviral development. A major obstacle commonly encountered, especially with respiratory viruses, is the emergence of virus mutational resistance, which stems from a commonly adopted strategy of directly targeting a specific part of a virus. Resistance of the N1 moiety of influenza A viruses to oseltamivir, a viral neuraminidase inhibitor, presents a growing concern [4,5]; there is also alarming concern over growing resistance to baloxavir marboxil, a recently introduced influenza antiviral that targets the viral PA protein [6]; and a mutation in the PB1 gene of influenza A virus readily confers resistance to favipiravir, a novel nucleoside analogue antiviral [7]. An alternative approach to antiviral design that can potentially circumvent the challenges of virus mutations is to regulate generic host innate immune responses during early infection to sufficiently disrupt virus replication to allow acquired immunity (antibody and cell-mediated) to be established. This novel host-centric antiviral approach can have the added benefit of inhibiting different types of viruses [8]. Given that acute respiratory virus infections of different aetiologies are clinically indistinguishable on presentation, a broad-spectrum host-centric antiviral that can target different virus types at the same time could significantly alter clinical management. 

Besides the present SARS-CoV-2 pandemic, influenza A virus and respiratory syncytial virus (RSV) continue to be important contributors to prevailing human respiratory viral infections. Influenza viruses are enveloped negative-sense segmented single-strand RNA viruses that globally cause up to 650,000 deaths each year and have been responsible for previous pandemics [9]. RSV is an enveloped negative-sense single-strand RNA virus and a leading cause of acute lower respiratory tract infection in young children and elderly people, associated with 48,000–74,500 in-hospital deaths in children aged under 5 years, with 99% of these deaths occurring in developing countries [10,11]. Whilst use of antivirals in influenza A virus infections is potentially hampered by virus resistance, use of the only approved chemical antiviral, ribavirin, against RSV is limited to severe infections in infants [5,12].

We recently showed that thapsigargin (TG), an inhibitor of the sarcoplasmic/endoplasmic reticulum (ER) Ca^2+^ ATPase pump [13], at nano-molar non-cytotoxic levels induces a potent host antiviral response that blocks influenza A virus replication [14]. The TG-induced ER stress unfolded protein response (UPR) appears as the primary driver responsible for activating a downstream spectrum of host antiviral defences. Here we show that TG is, in fact, a broad-spectrum host-centric inhibitor, highly effective in blocking the replication of RSV, common cold coronavirus OC43, SARS-CoV-2 and influenza A virus in immortalized or primary human cells. We also establish that TG is low pH-stable, as found in the stomach, able to rapidly induce a potent antiviral state lasting at least 48 h from a single priming, and therapeutically protective (in vivo) in mice, when orally administered post-infection, against a lethal influenza virus challenge.

## 2. Materials and Methods

### 2.1. Cells and Viruses

Primary normal human bronchial epithelial (NHBE) cells and bronchial epithelial growth media were supplied by Promocell (Heidelberg, Germany). Immortalised neonatal porcine tracheal epithelial (NPTr) cells [15], HEp2 cells, MRC5 cells, A549 cells, Calu-3 cells, Vero E6 cells and MDCK cells were cultured in DMEM-Glutamax supplemented with 10% foetal calf serum (FCS) and 100 U/mL penicillin–streptomycin (P/S) (Gibco, ThermoFisher Scientific, Paisley, UK). 

Human RSV (A2 strain, ATCC VR-1540), USSR H1N1 (A/USSR/77), PR8/H1N1 (A/PR8/1934) and pdm H1N1 virus (A/Swine/England/1353/2009 isolated through the DEFRA SwIV surveillance programme SV3401) were used in this study. Human coronavirus OC43 (OC43) and SARS-CoV-2 virus (strain 2019-nCoV/Italy-INMI1, clade V acquired from EVAg (008V-03893) were also used. The latter’s complete sequence was submitted to GenBank (SARS-CoV-2/INMI1-Isolate/2020/Italy: MT066156) and is available on GISAID website (betaCoV/Italy/INMI1-isl/2020: EPI_ISL_410545).

### 2.2. Cell Viability Assays

Cell viability was determined using a CellTiter-Glo Luminescent Cell Viability Assay kit or CellTiter-Glo 2.0 Cell Viability Assay (Promega, Madison, WI, USA), according to manufacturer’s instructions.

### 2.3. Chemical Priming of Cells

Stock TG (Merck, St. Louis, MO, USA) and remdesivir (Selleckchem, Munich, Germany) were dissolved in DMSO while ribavirin and hydroxychloroquine (Merck) were dissolved in water, as per the manufacturer’s recommendations. The antiviral regimen of the compounds had no detectable adverse effect on cell viability. Typically, unless otherwise described, cells were incubated in the indicated compound diluted in the appropriate cell culture medium (used within 30 min of dilution) for 30 min prior to infection, washed three times with PBS and infected with the indicated virus as described (pre-infection priming). Alternatively, cells were first infected for a specified period and then compound primed for 30 min followed by a continued infection culture (post-infection priming). All TG priming of cells was for a duration of 30 min only, followed by washing with PBS and downstream culturing according to specific experiments.

### 2.4. RSV Infection and Progeny Virus Quantification

HEp-2 cells, A549 cells and NHBE cells were infected at a specified multiplicity of infection (MOI) of RSV, based on 24 h plaque assays, in DMEM-Glutamax supplemented with 2% FCS for a total period of 48 or 72 h, and the spun supernatants collected for quantification by plaque assay or virus copy number by RT-qPCR (see Section 2.6). Titration of RSV by plaque assay on HEp2 cells has previously been described [16]. Briefly, HEp2 cells infected for 24 h with serial dilutions of the supernatants from infected cells (in triplicate) were fixed in acetone:methanol (1:1) and immunostained using mouse anti-RSV (2F7) antibody (1:1000) (Abcam, Cambridge, UK).

### 2.5. Influenza and Coronavirus Single and Co-Infection, and Progeny Virus Quantification

Influenza A virus infection requires serum-free medium supplemented with trypsin. The infection medium of the NPTr cells and A549 cells was Opti-MEM I (Gibco) supplemented with 100 U/mL P/S, 2 mM glutamine and 200 ng/mL L-1-tosylamide-2-phenylethyl chloromethyl ketone (TPCK) trypsin (Sigma-Aldrich). Cells were infected with a specified MOI of the influenza virus, based on focus forming assay (FFA), for 2 or 3 h in infection media, after which they were washed three times with PBS and incubated in fresh infection media for a period of 24 to 72 h, as indicated. Spun supernatants were collected for progeny virus quantification by 6 h FFA, tissue culture infectious dose 50% (TCID_50_) or virus copy number by RT-qPCR. Influenza virus titration by FFA has been previously described [14].

A549 cells and MRC5 cells were infected with a specified MOI of OC43, based on FFA, for 3 h in Opti-MEM I supplemented with 100 U/mL P/S, 2 mM glutamine and 100 ng/mL TPCK trypsin, after which they were washed three times with PBS and incubated in fresh infection media for a total of up to 72 h. A549 cells co-infected with the OC43 and USSR H1N1 virus were similarly treated. Spun supernatants were collected for progeny virus quantification by FFA or virus copy number by RT-qPCR. 

Calu-3, NHBE and Vero E6 cells were infected with SARS-CoV-2 at the indicated MOI, based on TCID_50_, for 3 h in Opti-MEM I supplemented with 100 U/mL P/S, 2 mM glutamine and 100 ng/mL TPCK trypsin, subsequently washed three times in PBS and incubated in fresh infection media for a total of up to 72 h, as indicated. Calu-3 cells co-infected with the pdm H1N1 virus and SARS-CoV-2 virus were similarly treated. Spun supernatants were collected for progeny virus quantification by TCID_50_ or virus copy number by RT-qPCR. TCID_50_ titrations for quantification of SARS-CoV-2 and influenza A virus were performed in the 96-well plate format, in quadruplicate as the minimum. Ten-fold dilutions were made of each sample in the growth media DMEM (Gibco). Then, 50 µL of diluted samples were added to MDCK and Vero E6 cells for influenza and SARS-CoV-2 titrations, respectively, and incubated at 37 °C for 1 h. The media was then replaced with 200 µL DMEM (supplemented with 200 ng/mL of TPCK trypsin for influenza titration) and incubated for 5 days at 37 °C. Each well was assessed for cytopathic effects under a microscope. Virus titres were calculated using the Spearman–Karber method as TCID_50_ [17,18]. The limit of detection was 5.62 TCID_50_/mL.

### 2.6. RNA Preparation and Real-Time RT-PCR

The RNeasy Plus Minikit (Qiagen, Hilden, Germany) was used to extract total RNA from cells. cDNA was synthesised from 1 µg of total RNA using the Superscript III First Strand synthesis kit (ThermoFisher Scientific). Expression of host genes was performed with a LightCycler−96 instrument (Roche, Basel, Switzerland). Computation of gene expression was based on the comparative Ct approach, normalised to *18S* ribosomal RNA. Human ER stress primers for *HSPA5* (FH1_HSPA5 and RH1_HSPA5), *HSP90B1* (FH1_HSP90B1 and RH1_HSP90B1) and *DDIT3* (FH1_DDIT3 and RH1-DDIT3); and human *IFNB1* primers (FH1_IFNB1 and RH1_IFNB1), *OAS1* primers (FH1_OAS1 and RH1_OAS1) and *RNASEL* primers (FH1_RNASEL1 and RH1_RNASEL1) were pre-made designs from Sigma-Aldrich. Additional primers (synthesised by Sigma-Aldrich) were designed using PrimerExpress 3.0.1 (ThermoFisher Scientific) and are shown in Table 1. The QIAamp Viral RNA Mini Kit (Qiagen) was used to extract viral RNA from the spun cell culture supernatants. One-step real-time qPCR was performed using the QIAGEN OneStep RT-PCR KIT to quantify relative virus copy number.

### 2.7. Western Blotting

Radioimmunoprecipitation assay (RIPA) buffer (Santa Cruz, Dallas, TX, USA), supplemented with 1% phenylmethylsulfonyl fluoride (PMSF) (Santa Cruz), 1% inhibitor cocktail and 1% sodium orthovanadate, was used to lyse the cells, and the protein concentration was measured by Bio-RAD protein assay (Bio-Rad, Hercules, CA, USA). Primary antibodies were goat anti-influenza A M1 at a 1:500 dilution (Abcam, ab20910), mouse anti-influenza A M2 clone 14C2 at 1:1000 (Invitrogen, MA1082), goat anti-influenza A virus polyclonal antibody at a 1:2000 dilution (Abcam, ab155877), mouse anti-coronavirus antibody OC-43 strain, clone 541–8F at 1:1000 (Sigma-Aldrich, MAB9012) and mouse anti-β-actin clone AC-74 at 1:5000 (Sigma-Aldrich, A2228). The appropriate species-specific secondary antibodies were peroxidase-conjugated (Abcam) for chemiluminescence detection (Amersham ECL Western Blotting Detection Reagent, Marlborough, MA, USA).

### 2.8. Interferon-β (IFNβ) ELISA

IFNβ secretion was quantified in the cell culture supernatant by enzyme-linked immunosorbent assay (ELISA). The IFNβ ELISA (Human IFN-beta Quantikine ELISA Kit, Bio-Techne, Minneapolis, MN, USA) was performed according to the manufacturer’s instructions. The samples were analysed in triplicate.

### 2.9. Influenza Virus Challenge in Mice

To assess the antiviral effectiveness of TG during infection (i.e., post-infection) and compare its performance with oseltamivir in vivo, 6- to 8-week-old BALB/c mice (female) were divided into two groups to determine post-infection survival (*n* = 8 per treatment group) and progeny virus production (*n* = 8 per treatment group). The mice were intranasally infected with 3 MLD_50_ of PR8/H1N1 virus. Twelve hours post-infection, TG (1.5 μg/kg/day), oseltamivir (45 mg/kg/day) or PBS+DMSO (percentage DMSO in PBS-DMSO control equal to other treatments) was orally administered by gavage daily for 5 days. At 3 and 5 days post-infection, the lungs of four mice from each group were collected for viral titration. Virus titration was performed by TCID_50_ assays on MDCK cells inoculated with 10-fold serially diluted homogenised lung tissues and incubated at 37 °C for 72 h. The TCID_50_ values were calculated according to the Reed–Muench method [19].

### 2.10. Quantification and Statistical Analysis

Statistical analysis was performed using GraphPad Prism 7 and the statistical method used described in the figure legend. A *p*-value < 0.05 was considered significant, and indicated as * *p* < 0.05, ** *p* < 0.01, *** *p* < 0.001 and **** *p* < 0.0001. The Kaplan–Meier method was employed for the survival analysis. The presented results are representative of three or more independent repeats, and the error bars, unless otherwise stated, are standard deviations. 

### 2.11. Ethics Statement

All animal work was approved by the Beijing Association for Science and Technology (ID: SYXK (Beijing) 2007–0023) and conducted in accordance with the Beijing Laboratory Animal Welfare and Ethics guidelines, as issued by the Beijing Administration Committee of Laboratory Animals, and in accordance with the China Agricultural University Institutional Animal Care and Use Committee guidelines (ID: SKLAB-B-2010–003).

## 3. Results

### 3.1. TG Blocks Progeny RSV Production

To demonstrate the antiviral activity of TG against RSV, human HEp2 and A549 cells were briefly primed with TG for 30 min at either 24 h before infection (Figure 1A) or 24 h post-infection (hpi) (Figure 1B). Pre-infection and post-infection TG priming of each cell type resulted in dramatic, statistically significant, reduction in progeny virus production. HEp2 cells primed with 0.5 µM TG before infection reduced progeny virus production by almost 10,000-fold (Figure 1A); cells primed with 0.5 µM TG at 24 hpi reduced virus output by approximately 1000-fold (Figure 1B), which respectively suggests its prophylactic and therapeutic antiviral potential against RSV. A single 30-min priming of HEp2 and A549 cells with TG induced a potent antiviral state that lasted at least 48 h (Figure 1C,D). TG priming of HEp2 cells (immediately before infection or 48 h before infection) targeted transcriptional inhibition of the viral L, F and M genes (Figure 1E,F). The antiviral regimen of TG used was non-cytotoxic in HEp2 and A549 cells (Figure 1G,H). Additional evidence of non-cytotoxic antiviral dose regimen of TG is found in our earlier publication [14]. Thus, TG as a non-cytotoxic inhibitor was rapid-acting, induced an antiviral state in cells lasting at least 48 h in duration, and was effective when used before or during active RSV infection.

Next, the antiviral performance of TG was compared with ribavirin, an approved antiviral for use on young children with severe RSV infection, according to progeny virus output (Figure 2A) and viral RNA detection (Figure 2B) in media of infected HEp2 cells. TG was superior to ribavirin in blocking RSV replication. For instance, pre-infection priming of HEp2 cells with 0.1 µM TG for 30 min was 160-fold more inhibitory to progeny virus production than the continuous use of ribavirin at 30 µM (ribavirin effective concentration at 50% (EC_50_) against RSV = 11 µM [20]) (Figure 2A). TG exhibited a selectivity index (cytotoxic concentration at 50% (CC_50_)/EC_50_) of 984 and EC_90_ of 84.55 nM in RSV inhibition in HEp2 cells, suggesting a strong safety margin (Figure 2C,D). In TG-primed and infected primary NHBE cells, reduced viral RNA detection in the corresponding media (Figure 2E) was coincident with transcriptional inhibition of viral L, F and M genes (Figure 2F), which was further manifested as a TG dose-dependent reduction in viral protein production (Figure 2G,H). Taken together, TG was more effective than ribavirin as an antiviral, showed a strong selectivity index and blocked RSV virus transcription and viral protein production.

In primary NHBE cells, pre-infection priming with TG enhanced expression of the ER stress genes (*DDIT3*, *HSPA5* and *HSP90B1*), relative to the DMSO control, before and during RSV infection in a TG dose-dependent manner (Figure 3A,C). TG also increased the basal expression of the RIG-I signalling-associated genes (*RIG-I*, *IFNB* and *RNASEL*) in NHBE cells, but during infection, the RIG-I-associated gene induction, relative to the infected DMSO control, was noticeably attenuated (Figure 3D,G). Thus, the reduced transcriptional induction of the RIG-I-associated genes during RSV infection in NHBE cells was a feature of TG-mediated RSV inhibition.

### 3.2. TG Blocks Progeny Coronavirus OC43 Production

To demonstrate TG inhibition of coronaviruses, A549 cells were primed with TG for 30 min immediately prior to infection with endemic OC43 virus [21] (Figure 4A–C). In a dose dependent manner, TG blocked virus replication, as evidenced by the sharp reduction in viral RNA in the media of infected cells (Figure 4A), inhibition in viral transcription (Figure 4B) and reduced viral NP protein production (Figure 4C). The antiviral performance of TG was compared with hydroxychloroquine (HC), a less toxic compound than chloroquine shown to inhibit OC43 [22]. TG was much more effective than HC in blocking the production of progeny OC43 virus (Figure 4D,E). Pre-infection priming with 0.05 µM TG for only 30 min was more inhibitory to OC43 than the continuous use of HC at 20 µM throughout infection (HC EC_50_ against SARS-CoV-2 = 4.51 µM [23]) (Figure 4D). Unlike the modest reduction in progeny virus from HC treatment, hardly any virus was detected in the media of the TG-primed infected cells (Figure 4E). In A549 cells, TG was significantly better able than remdesivir (RDV) [24], a nucleoside analogue recently approved for emergency use in SARS-CoV-2 infection, at blocking OC43 virus (Figure 4F) and influenza virus (Figure 4G) replication. Although, at 72 hpi, cells continuously exposed to 0.3 µM RDV (RDV EC_50_ against OC43 = 0.15 µM [24]) showed around a 17,000-fold reduction in progeny viral RNA detection (relative to corresponding DMSO control); cells primed with 0.3 µM TG, in turn, was more inhibitory than the RDV-treated cells by 450-fold (Figure 4F). At 72 hpi, 0.05 µM TG-primed cells also inhibited USSR H1N1 virus production more than the continuous use of RDV at 0.3 µM by 7-fold; RDV had little antiviral effect on influenza virus replication (Figure 4G). The antiviral use of TG in A549 cells was non-cytotoxic (Figure 4H). The selectivity index (CC_50_/EC_50_) of TG on OC43 in MRC5 cells was high at between 7072 and 9227 (Figure 4I). Collectively, TG strongly inhibited OC43 virus transcription and protein production, was more effective as an antiviral than HC and RDV and exhibited a high selectivity index.

Next, the ER stress response to TG priming before and during OC43 virus infection was assessed. In A549 cells, TG in a dose dependent manner stimulated the expression of the ER stress genes basally and during infection. The TG-induced ER stress gene profiles from OC43 infection of the A549 cells (Figure 5A,C) were highly similar to those from RSV infection of NHBE cells (Figure 3A,C). In A549 cells, TG priming appeared to have little effect on the basal transcription of the RIG-I-associated genes (*RIG-I*, *IFNB* and *OAS1*). Like in RSV infection of the NHBE cells (Figure 3D,G), induction of the RIG-I-associated genes was also attenuated during OC43 infection of the TG-primed cells (Figure 5D,F). Thus, reduced transcriptional induction of the RIG-I-associated genes during OC43 infection was also a feature of the TG-mediated OC43 virus inhibition. Importantly, pre-infection TG-primed A549 cells were just as able to inhibit separate virus infection (Figure 5G) as co-infection (Figure 5H) with the OC43 virus and USSR H1N1 influenza A virus. In summary, TG priming of A549 cells increased the expression of the ER stress genes basally and during OC43 infection, attenuated induction of the RIG-I signalling-associated genes during infection, and inhibited co-infection with the OC43 and influenza viruses.

### 3.3. TG Blocks Progeny SARS-CoV-2 Production

SARS-CoV-2 was as susceptible to TG inhibition as the OC43 virus. Pre-infection priming of Calu-3 and NHBE cells with TG blocked SARS-CoV-2 replication (Figure 6A,C), which is comparable to the inhibition seen with the OC43 virus in A549 and MRC5 cells (Figure 4A,D). In Calu-3 cells, post-infection priming with TG for 30 min at 24 hpi with SARS-CoV-2 was also effective in virus inhibition, indicating its therapeutic potential in SARS-CoV-2 infection (Figure 6B). As with influenza virus inhibition [14], TG was unable to inhibit the replication of SARS-CoV-2 in Vero E6 cells, suggesting that an intact type I IFN system is necessary for TG-mediated host antiviral response (Figure 6D). 

The broad-spectrum antiviral potency of TG was just as evident in separate virus infections as in the co-infection with the SARS-CoV-2 and pdm H1N1 viruses, as determined by progeny viral RNA detection (Figure 6E–H) and infectious progeny virus detection by TCID_50_ assays (Figure 6I–K) in the culture media of infected Calu-3 cells. At 72 hpi, 0.5 µM TG-primed cells inhibited production of the SARS-CoV-2 progeny virus by 300-fold (99.7%) and 880-fold (99.9%), relative to the corresponding DMSO controls, in single-virus infection (Figure 6I) and co-infection (Figure 6J), respectively. At 72 hpi of co-infected cells primed with 0.5 µM TG, production of the pdm H1N1 virus was reduced 11-fold (93.3%) (Figure 6K) relative to the corresponding DMSO control. Overall, reduction in progeny virus from TG priming was proportionally higher with SARS-CoV-2 than pdm H1N1 virus infection in single-virus comparison (Figure 6E,F) and co-infection (Figure 6G,H), suggesting that SARS-CoV-2 was more sensitive to TG inhibition than the pdm H1N1 virus. In summary, TG as a broad-spectrum antiviral targeted the major human respiratory viruses of RSV, coronaviruses (in particular SARS-CoV-2) and influenza A viruses, and did not distinguish between single-virus and combined-virus infections.

### 3.4. TG Post-Translational Block of Influenza A Virus Therapeutically Protects Mice in Lethal Virus Challenge

We previously found that TG inhibition of influenza virus replication in NHBE cells and NPTr cells was accompanied by little or no change in viral transcription or production in viral NP and M1 proteins, indicating that the virus was blocked post-translationally [14]. As viral NP and M1 proteins are translated on cytosolic ribosomes for nuclear import, we examined here the viral proteins (HA, NA and M2) that are processed through the ER-Golgi apparatus destined for the host cell membrane [25]. Viral protein analysis from TG-primed NPTr cells (Figure 7A,B) showed that expression of all the additional viral proteins examined appeared also to be unaffected, which indicates that TG differentially targets the replication cycle of the influenza virus, and RSV and coronavirus. As the antiviral effects of TG appeared stable in an acidic pH, but not an alkaline pH (Figure 7C,D), the oral therapeutic (post-infection) efficacy of TG was assessed in a lethal influenza virus challenge in mice. Each BALB/c mouse (*n* = 8 per group) in a group was first intranasally infected with 3 MLD_50_ of the PR8/H1N1 virus and the next day was dosed with TG or oseltamivir by gavage once daily for 5 days. Protection conferred by low oral dose TG (empirically chosen at 1.5 μg/kg/day) was similar to that of high dose oseltamivir (45 mg/kg/day) in mice. Therapeutically, the TG-treated group showed significantly improved survival (Figure 7E), reduced virus shedding at 3 and 5 days post-infection (dpi) (Figure 7F) and less severe weight loss (Figure 7G,H) relative to the PBS-DMSO control. Seven, all eight and two mice survived in the TG, oseltamivir and PBS+DMSO group, respectively. There was no significant difference between the TG- and oseltamivir-treated mice in all three parameters. Thus, oral TG conferred therapeutic protection in mice subjected to a lethal influenza virus challenge.

## 4. Discussion

We recently reported that TG at non-cytotoxic levels induces a potent host antiviral response that blocks influenza A virus replication [14]. Here we further demonstrated that TG is an acid-stable, broad-spectrum inhibitor that is highly effective against RSV, coronavirus (OC43 and SARS-CoV-2) and influenza A virus. The three major RNA virus types represent diverse viral genomic makeup and life cycles. Here, the antiviral potency of TG is a physiological response to reversible ER stress activation, and is in stark contrast to the early experimental use of TG (at high µM concentrations over extended periods) as an inducer of apoptosis, which is an end-stage outcome of extreme ER stress. All accumulated evidence to date indicates that the antiviral regimen of TG is non-cytotoxic, showing high selectivity indices against RSV, OC43 virus and influenza virus, and is devoid of any overt adverse effects in mice. Previous work found no virus resistance arising from cell-passages of influenza A virus in continuous sub-optimal presence of TG [14]. The lasting antiviral state of at least 48 h from short TG exposure, along with its ability to inhibit different viruses before or during active infection, makes TG (or its derivatives) attractive as a prophylactic and therapeutic antiviral candidate. Additionally, TG’s antiviral performance was significantly better than RDV and ribavirin in their respective inhibition of coronavirus OC43 and RSV. A further advantage of TG as an antiviral is that it is just as inhibitory to coronaviruses (OC43 and SARS-CoV-2) and influenza viruses (USSR H1N1 and pdm H1N1) in separate infections as well as in co-infections. One of the major problems of influenza management is either the need to confirm infection by laboratory or rapid diagnostic test prior to instigating therapy; or to treat empirically and likely use influenza-specific therapy against non-influenza pathogens. A broad-spectrum antiviral such as TG may, in time, change this paradigm.

In the control of highly infectious diseases, effective vaccines and antivirals are necessary. In the present COVID-19 pandemic, vaccines have started to become available for mass vaccination. There is, however, no approved antiviral against COVID-19 for prophylaxis or early-infection community use in the general population. The WHO Solidarity Trial [26] found none of the antivirals examined (RDV, HC and IFNB1a) to have any significant impact on mortality or duration of hospitalisation. Other drugs in current use for the treatment of COVID-19, such as dexamethasone and hydrocortisone, are symptomatic immune modifiers that modulate the damaging effects of the virus-induced “cytokine storm” and are not antivirals. Thus, effective antivirals that can intervene during an active infection, especially at the initial stages of the disease, are very much needed, even with the availability of vaccines. Given the emergence of coronavirus mutants, which may undermine vaccines currently in use and under development, and other major respiratory viruses co-circulating in populations, effective antivirals that do not discriminate between virus types are all the more necessary to manage current and future pandemics. The enormous potential practical benefits of TG over current antivirals can be envisaged in general practice, if such an orally administered broad-spectrum antiviral could be dispensed for early intervention without the need to establish a specific respiratory virus diagnosis; such an antiviral will fulfil an integral role with vaccines in reducing the public health burden of SARS-CoV-2 and other respiratory viruses.

We previously established that TG-induced ER stress leading to UPR is a key innate immune driver that mediates a range of host-centric antiviral processes to block influenza A virus replication [14]. In a recent pre-print, TG was reported to efficiently inhibit coronavirus (hCoV-229E, MERS-CoV, SARS-CoV-2) replication at concentrations below its cytotoxic range [27]. The mechanistic antiviral basis arising from the primary ER stress response, however, remains to be fully elucidated. Noteworthy is that TG treatment triggers a downstream spectrum of generic host antiviral defences that are remarkably effective in blocking the replication of different RNA viruses. TG priming consistently increased the basal transcriptional expression of the ER stress genes (*DDIT3*, *HSPA5* and *HSP90B1*) in a dose-dependent manner. Infection with the RSV and OC43 virus of the TG-primed cells further raised the ER stress gene expression but attenuated the induction of the RIG-I-associated genes. For both virus types, TG inhibition of progeny virus production was accompanied by a reduced viral transcription and viral protein expression. In contrast, as evidenced by no change in the detection of all viral proteins examined, TG targeted influenza virus replication post-translationally, possibly involving post-translational viral protein modifications and transport. Influenza virus infection of the TG-primed cells also attenuated the ER stress gene response but enhanced expression of the RIG-I-associated genes [14]. In summary, TG targets respiratory viruses at different stages of viral replication; RSV and coronavirus OC43 are inhibited from the level of transcription whereas influenza A virus is targeted post-translationally.

Differences in host ER stress and RIG-I-associated responses, and differences in virus-stage inhibition between infections with RSV and the OC43 virus, and influenza virus in TG-primed cells, indicate that there are functional redundancies in the multiple antiviral processes activated by TG to control different viruses at different stages of their life cycles. Thus, in TG-primed cells, transcriptional ER stress response to infection is virus-dependent. RSV and OC43 virus infection elicit strong ER stress gene expression in TG-primed cells, which, in turn, correlates with attenuated RIG-I-associated transcriptional activation; conversely, influenza virus infection attenuated the transcriptional activation of the ER stress genes in TG-primed cells, which is associated with elevated RIG-I-associated transcriptional activation. Notably, a functional type I IFN system appears essential in TG–ER stress-mediated inhibition of SARS-CoV-2 replication and influenza A virus replication [14]. Given that in TG-primed cells the RIG-I-associated response differed between RNA virus types, the qualitative response (such as temporal regulation) of this antiviral signalling pathway could be more important than the mere intensity of the response in inhibiting a specific virus replication. Regardless of the manner of ER stress response to infection, TG is manifestly highly protective against all three major virus types. In conclusion, we propose that TG (or its derivatives) is a promising orally active, broad-spectrum antiviral against SARS-CoV-2, RSV and influenza virus, and has the prospect to be a defender against the next Disease X pandemic.

## 5. Patents

The use of TG and other structurally-related compounds in antiviral therapy is covered by patent PCT/GB2019/050977 and PCT/GB2020/052479.

## Figures and Tables

**Figure 1 viruses-13-00234-f001:**
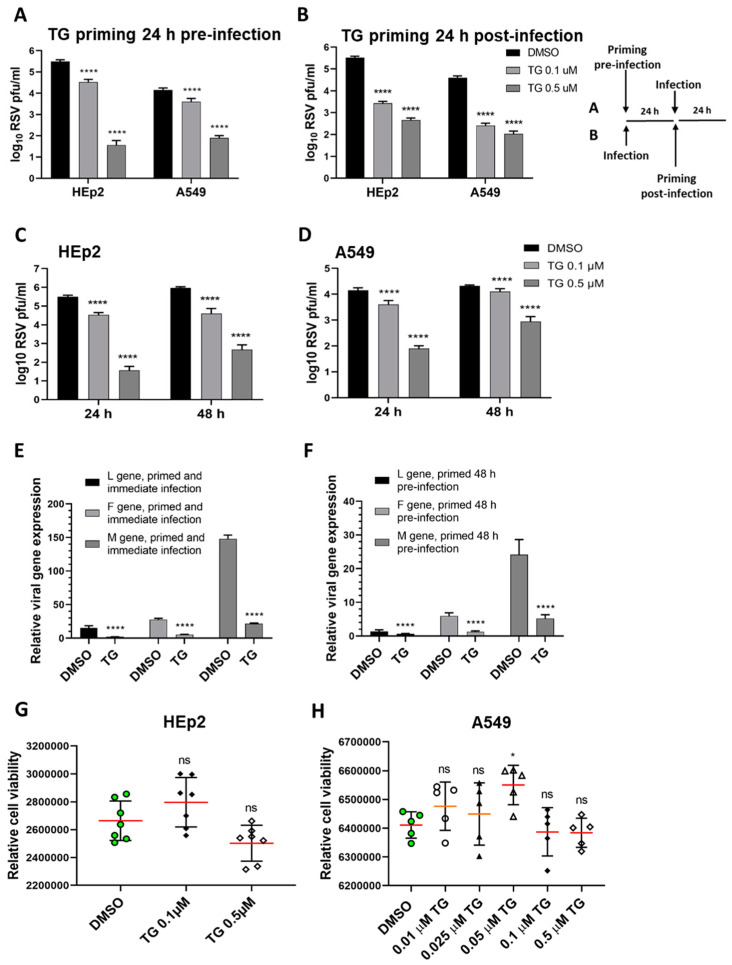
Short exposure (30 min) of thapsigargin (TG) to human cells, at non-cytotoxic levels, promptly elicits a prolonged (≥48 h) antiviral state that blocks RSV replication. TG priming 24 h before (**A**) or 24 h after (**B**) infection, in a dose-dependent manner, blocks RSV production. (**A**) HEp2 and A549 were primed with TG or control DMSO for 30 min, washed with PBS and incubated in normal culture media for 24 h before RSV infection, or (**B**) were initially infected with RSV for 24 h before 30 min of TG priming. The TG-induced antiviral state lasted at least 48 h. HEp2 (**C**) and A549 (**D**) cells were primed with TG or DMSO control for 30 min, as indicated, washed with PBS and allowed a further period of 24 or 48 h of normal culture; after which cells were infected with RSV. All cells were infected with RSV (A2 strain, ATCC VR-1540) at 0.1 MOI for a total duration of 3 days. The spun supernatants were collected to infect HEp2 cells for 24 h for immuno-detection of RSV with mouse anti-RSV (2F7) antibody (pfu/mL). Significance by 2-way ANOVA (Sidak’s multiple comparisons) is relative to the corresponding DMSO control. HEp2 cells were more permissive to RSV replication than A549 cells. TG inhibited RSV transcription. HEp2 cells were (**E**) primed for 30 min with 0.5 µM TG immediately before RSV infection, and (**F**) primed with TG for 30 min, washed with PBS and cultured for a further period of 48 h before RSV infection. After a total of 3 days of infection, total RNA was extracted for cDNA conversion to quantify viral gene (L-gene, M-gene and F-gene) expression normalised to *18s* rRNA. Significance was determined by unpaired *t*-test relative to expression in the corresponding DMSO control. (**G**) HEp2 and (**H**) A549 cells pre-incubated with indicated concentrations of TG or DMSO control for 30 min were washed with PBS, cultured overnight in serum-free media (Opti-MEM) and subjected to cell viability assays (CellTiter-Glo^®^ Luminescent Cell Viability Assay kit, Promega). Horizontal bars = mean (red) ± standard deviation; ns = not significant. Significance by one-way ANOVA with Dunnett’s multiple comparisons is relative to the corresponding DMSO control. All assays were in triplicates and were performed three times. * *p* < 0.05 and **** *p* < 0.0001.

**Figure 2 viruses-13-00234-f002:**
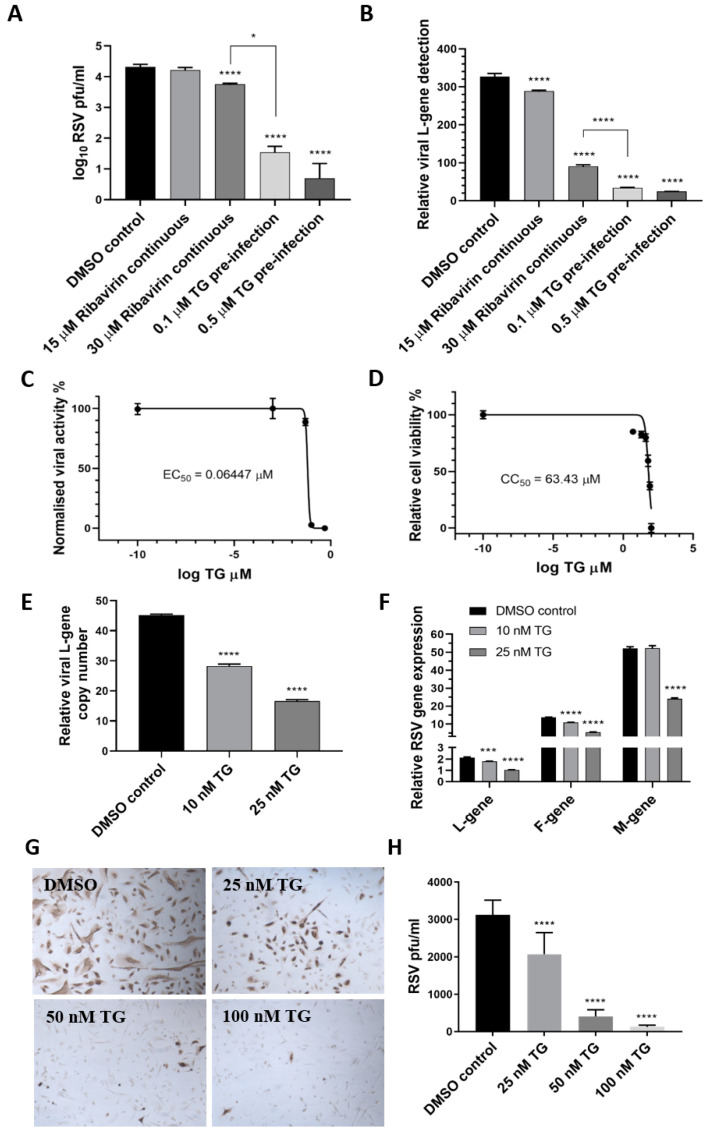
TG is more effective as an antiviral than ribavirin, shows a high selectivity index and blocks viral transcription and viral protein production. HEp2 cells were primed with TG, ribavirin or DMSO control, as indicated, for 30 min, washed with PBS and infected with RSV, as earlier described, in the continuous presence of ribavirin in media or in media alone (for TG or DMSO control). Media were harvested at 4 dpi for (**A**) progeny virus titration (pfu/mL) and (**B**) detection of viral L-gene RNA by one-step reverse transcription-qPCR. Unless otherwise indicated, significance by one-way ANOVA (Tukey’s multiple comparisons) is relative to the DMSO control. The corresponding selectivity index (SI) of TG in RSV infection of HEp2 cells is estimated at 984 (**C**,**D**). The effective concentration (EC) and cytotoxic concentration (CC) of TG against RSV in HEp2 cells were determined by pfu/mL virus titrations and luminescence cell viability assays, respectively, over a pre-infection priming range of TG. SI = CC_50_/EC_50_ = 63.43/0.06447 = 983.9. EC_90_ = 84.55 nM. NHBE cells were primed with TG or the DMSO control, as indicated, for 30 min, washed with PBS and infected with RSV at 0.1 MOI. At 48 hpi, media were harvested for the detection of the viral L-gene RNA by one-step reverse transcription-qPCR (**E**), and total RNA was extracted for cDNA conversion to quantify the expression of viral genes (L-gene, F-gene and M-gene) normalised to *18s* rRNA (**F**). Significance by one-way ANOVA (Dunnett’s multiple comparisons) (**E**) and by 2-way ANOVA (Tukey’s multiple comparisons) (**F**) is relative to the DMSO control. TG also inhibited viral F-protein production in NHBE cells. Cells were primed with TG or the DMSO control, as indicated, for 30 min, washed with PBS and infected with RSV at 0.05 MOI for 48 h and directly immunostained for the presence of RSV F-protein (**G**,**H**). Images captured at 100 times magnification. There was a clear reduction in the number of RSV-positive cells (pfu) with an increasing priming dose of TG. Significance by one-way ANOVA (Dunnett’s multiple comparisons) is relative to the DMSO control. * *p* < 0.05, *** *p* < 0.001 and **** *p* < 0.0001.

**Figure 3 viruses-13-00234-f003:**
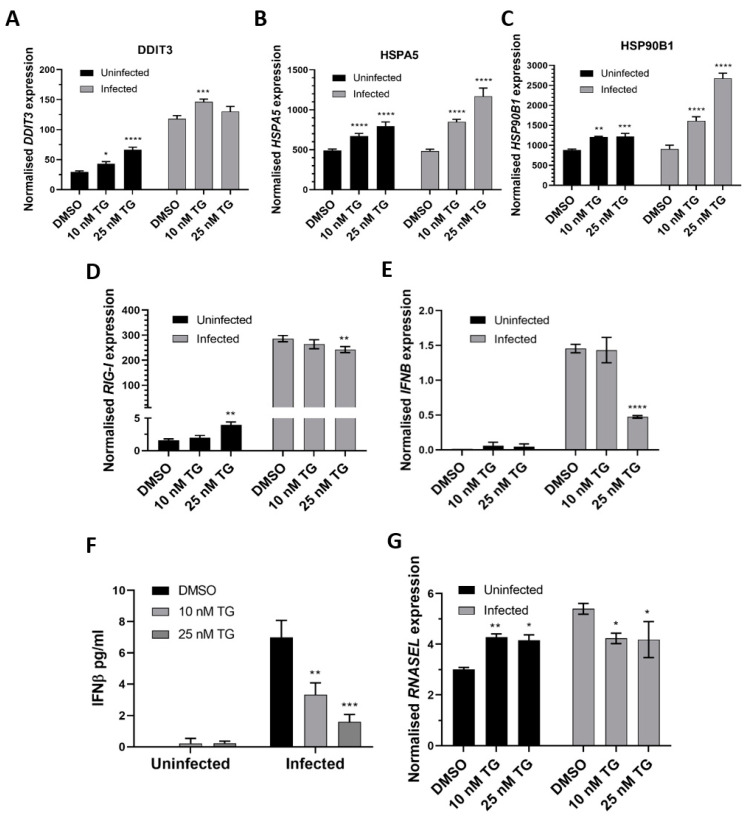
TG priming of NHBE cells increases expression of the ER stress genes (pre- and post-infection) and basal expression of the RIG-I signalling-associated genes, but during infection, induction of the RIG-I-associated genes is attenuated. Cells were primed with TG or DMSO control for 30 min, washed with PBS and infected with RSV at 0.05 MOI 48 h after which total RNA was extracted for cDNA conversion to quantify expression of the ER stress genes (**A**) *DDIT3*, (**B**) *HSPA5* and (**C**) *HSP90B1*. Expression was normalised to *18s* rRNA and the indicated significance, determined by 2-way ANOVA (Sidak’s multiple comparisons), is relative to the corresponding DMSO control. There was consistent indication of pre-infection activation of the RIG-I-associated genes, *RIG-I* (**D**), *IFNB* (**E**) (and corresponding IFNB protein, (**F**)) and *RNASEL* (**G**) from TG priming. IFNB ELISA was performed on supernatants at 16 hpi. All RNA expression was normalised to *18s* rRNA and the indicated significance, determined by one-way ANOVA and Dunnett’s multiple comparisons (**D**), or 2-way ANOVA Tukey’s multiple comparisons (**E**–**G**), is relative to the corresponding DMSO control. All assays were in triplicates and were performed three times. * *p* < 0.05, ** *p* < 0.01, *** *p* < 0.001 and **** *p* < 0.0001.

**Figure 4 viruses-13-00234-f004:**
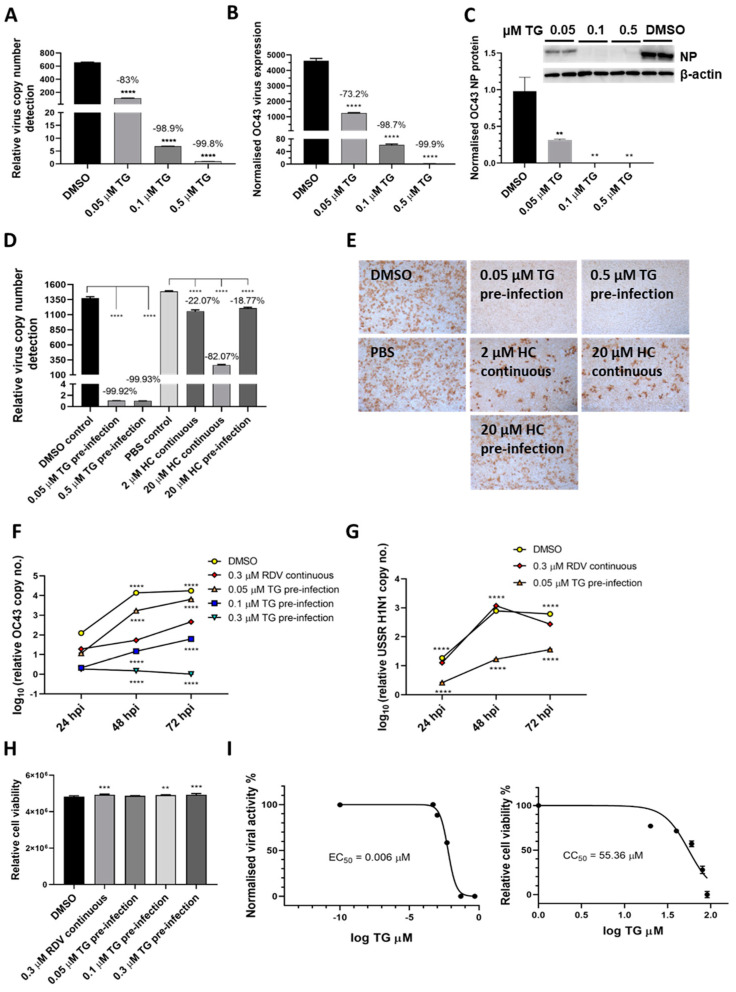
TG inhibits OC43 viral transcription and protein production, is more effective as an antiviral than hydroxychloroquine and remdesivir, and exhibits a high selectivity index. (**A**,**B**) A549 cells were primed with TG for 30 min, washed twice with PBS and infected with OC43 at 0.5 MOI for 3 h; cells were then washed again with PBS and cultured for 48 h in serum-free media (Opti-MEM supplemented with 0.1 µg/mL TPCK trypsin), after which RNA was extracted from (A) culture media and (**B**) cells for one-step reverse transcription qPCR, and cDNA conversion followed by qPCR, respectively, to detect viral OC43 replicase polyprotein 1ab RNA; cDNA quantification was normalised to *18s* rRNA. (**C**) Duplicate set of wells were used for cellular protein extractions to detect OC43 NP by Western blotting. (**D**,**E**) MRC5 cells were primed with TG, hydroxychloroquine (HC) or the DMSO/PBS control, as indicated, for 30 min, washed twice with PBS, infected with OC43 (at 0.01 MOI) for 3 h, further washed with PBS twice and finally replenished with serum-free media in the absence of a compound (pre-infection) or continued presence of HC (continuous). At 2 dpi, media were collected for (**D**) detection of viral polyprotein 1ab RNA by one-step reverse transcription-qPCR and (**E**) direct progeny virus detection by infecting A549 cells for 24 h and immunostaining for the presence viral NP. Images captured at 100 times magnification. The indicated significance (determined by one-way ANOVA) and percentage reduction in viral RNA detection are relative to the corresponding controls. (**F**,**G**) TG was superior to remdesivir (RDV) in blocking OC43 (**F**) and influenza A virus (**G**) replication. The A549 cells were primed with the indicated TG, 0.3 µM RDV or DMSO control for 30 min, washed twice with PBS and infected with 0.01 MOI of CoV OC43 or 1.0 MOI of USSR H1N1 virus for 2 h, after which cells were washed again with PBS and incubated in serum-free media for TG primed cells, or in media in the continuous presence of RDV. At 24, 48 and 72 hpi, viral RNA extraction was performed on the collected supernatants followed by one-step reverse transcription qPCR to detect the relative copy number of OC43 replicase polyprotein 1ab RNA or influenza M-gene RNA, based on the relative Ct method. The indicated significance is relative to the corresponding RDV-treated cells based on 2-way ANOVA Dunnett’s (**F**) or Tukey’s (**G**) multiple comparisons test. (**H**) RDV and TG treatments had no adverse effect on cell viability. A549 cells were treated continuously with RDV, or for 30 min with the indicated TG or DMSO, washed, cultured overnight and subjected to cell viability assay (CellTiter-Glo 2.0 Cell Viability Assay, Promega). The indicated significance was determined by one-way ANOVA relative to the DMSO control. (**I**) Selectivity index (CC_50_/EC_50_) of TG in OC43 inhibition was estimated at between 7072 and 9227. EC_90_ = 0.02622 μM. MRC5 cells were primed with TG (0 to 91 µM) for 30 min, washed twice with PBS and culture in DMEM Glutamax with 10% FCS and 1% P/S overnight. Cell viability assay (CC_50_) was performed with CellTiter-Glo 2.0 Cell Viability Assay (Promega). The effective or inhibition TG dose response (EC_50_) was based on priming of MRC5 cells with the indicated concentrations of TG (0 to 0.5 µM) for 30 min followed by PBS washing and infection with OC43 at 0.01 MOI. Three days post-infection, the supernatants were harvested for RNA extraction and one-step reverse-transcription qPCR to quantify the presence of viral RNA (polyprotein 1ab RNA). CC = cell cytotoxicity; EC = effective concentration. ** *p* < 0.01, *** *p* < 0.001 and **** *p* < 0.0001.

**Figure 5 viruses-13-00234-f005:**
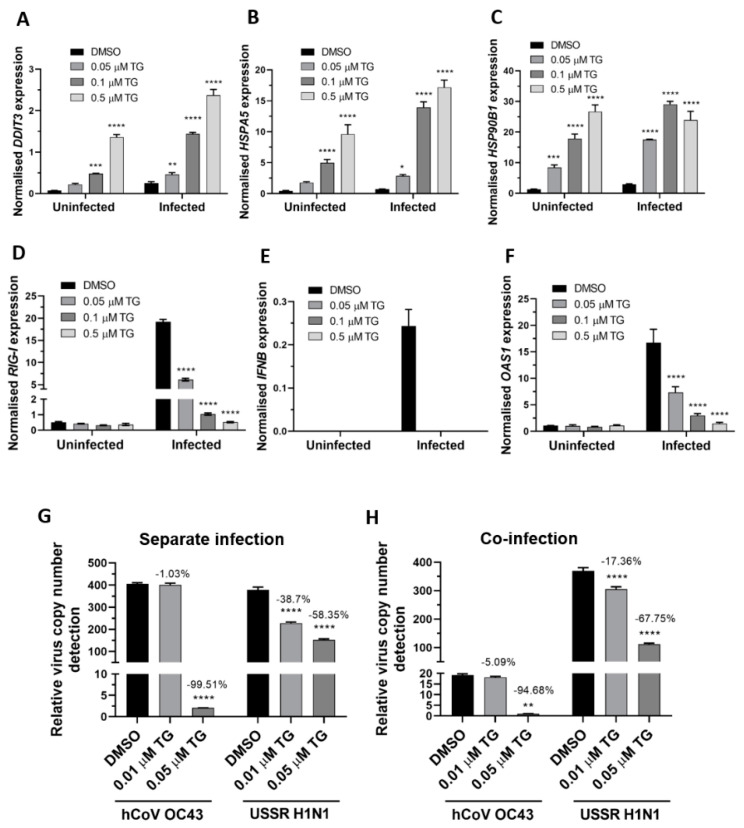
TG priming of A549 cells increases expression of ER stress genes basally and during OC43 infection, attenuates induction of the RIG-I signalling-associated genes during infection, and inhibits co-infection with the OC43 and influenza viruses. (**A**–**C**). TG priming appeared to stimulate ER stress gene expression in a dose-dependent manner. (**D**–**F**) TG attenuated the induction of the RIG-I-associated genes during infection. A549 cells were primed with TG for 30 min, washed twice with PBS and infected with OC43 at 0.5 MOI for 3 h; thereafter cells were washed again with PBS and cultured for 24 h in serum-free media, after which cell lysates were harvested for RNA extraction and cDNA conversion for qPCR of (**A**) *DDIT3*, (**B**) *HSPA5*, (**C**) *HSP90B1*, (**D**) *RIG-I*, (**E**) *IFNB* and (**F**) *OAS1*. All expression was normalised to *18s* rRNA. Significance is relative to the corresponding DMSO control based on 2-way ANOVA (Tukey’s multiple comparisons). (**G**,**H**) In A549 cells, TG inhibited the replication of OC43 virus and USSR H1N1 virus in separate virus infection or in co-infection. Cells were primed with TG for 30 min, washed twice with PBS and infected with the OC43 virus and USSR H1N1 virus at 0.01 and 1.5 MOI, respectively (based on FFAs), as single virus infection or co-infection for 3 h; after which cells were again washed twice with PBS and incubated in serum-free media. Culture media were harvested at 48 hpi for viral RNA extraction followed by one-step reverse transcription qPCR to detect the relative copy number of OC43 replicase polyprotein 1ab RNA and USSR H1N1 M-gene RNA. The indicated significance is based on 2-way ANOVA Tukey’s multiple comparisons and the percentage reduction in viral RNA detection are relative to the corresponding DMSO control. All assays were in triplicates and were performed three times. * *p* < 0.05, ** *p* < 0.01, *** *p* < 0.001 and **** *p* < 0.0001.

**Figure 6 viruses-13-00234-f006:**
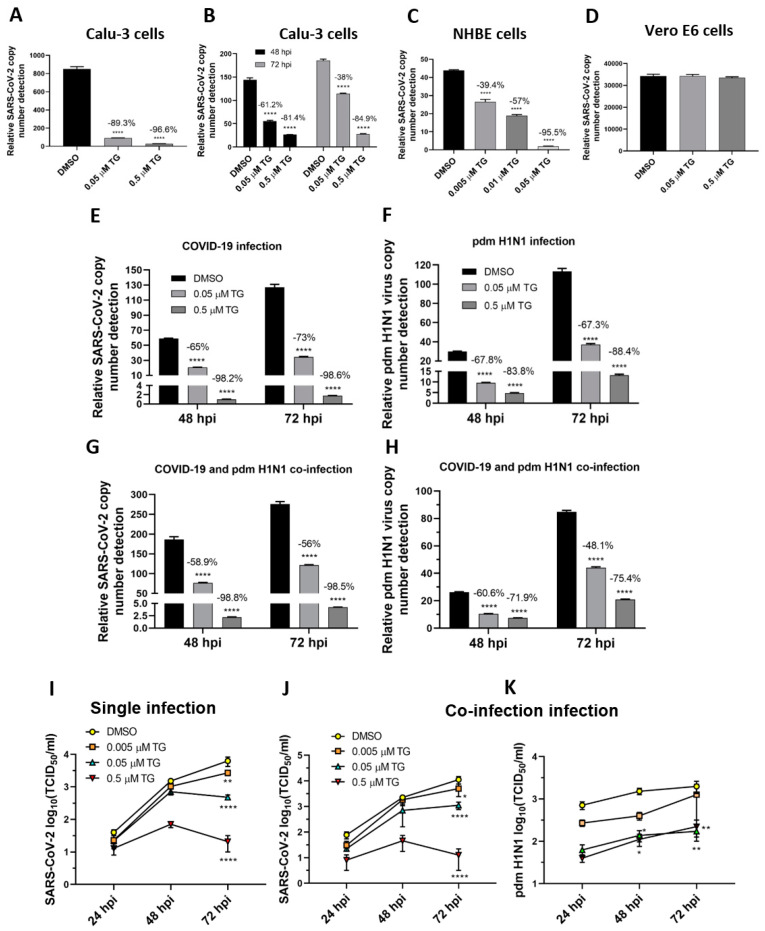
TG effectively blocks progeny SARS-CoV-2 production in single-virus infection and in co-infection with the pdm H1N1 virus. (**A**,**C**,**D**) Pre-infection TG priming of Calu-3 and NHBE cells, but not Vero E6 cells, effectively inhibited progeny virus output. Cells were primed with TG as indicated for 30 min, washed twice with PBS and infected with SARS-CoV-2 at 0.01 MOI for 3 h; after which cells were again washed twice with PBS and incubated in serum-free media, supplemented with 0.2 µg/mL TPCK trypsin. Viral RNA extraction was performed on culture media at 72 hpi. (**B**) Priming with TG at 24 hpi blocked replication of SARS-CoV-2 in Calu-3 cells. Cells were first infected with SARS-CoV-2 at 0.01 MOI for 24 h, then primed with the indicated TG for 30 min, washed 3 times with PBS and incubated in serum-free media. Viral RNA extraction was performed on culture media at 48 and 72 hpi. All RNA isolated above were subjected to one-step reverse transcription qPCR to detect the relative copy number of SARS-CoV-2 replicase polyprotein 1ab RNA, based on the relative Ct method. Pre-infection priming of Calu-3 cells with TG inhibited separate infection of (**E**) SARS-CoV-2 and (**F**) the pdm H1N1 virus, and (**G**,**H**) co-infection with both viruses. Cells were primed with TG or DMSO for 30 min, infected with the corresponding virus at 0.01 MOI for 2 h, washed 3 times with PBS and incubated in serum-free media. At indicated time points post-infection, the media of infected cells were sampled for viral RNA extraction to perform one-step reverse transcription qPCR to detect the relative copy number of SARS-CoV-2 replicase polyprotein 1ab RNA (**E**,**G**) and influenza M-gene RNA (**F**,**H**). (**I**–**K**) Media samples taken at 24, 48 and 72 hpi from similarly infected cultures were used to detect viable progeny virus by TCID_50_ virus titration in Vero cells (displayed as mean ± SEM). (**I**) In single-virus infection with SARS-CoV-2, TG exhibited dose-dependent virus inhibition. In co-infection with SARS-CoV-2 and pdm H1N1 virus, TG was just as capable in inhibiting SARS-CoV-2 (**J**) and pdm H1N1 virus (**K**) at the same time. The indicated significance is based on 2-way ANOVA Tukey’s multiple comparisons test and percentage viral RNA change relative to the corresponding DMSO control. * *p* < 0.05, ** *p* < 0.01 and **** *p* < 0.0001.

**Figure 7 viruses-13-00234-f007:**
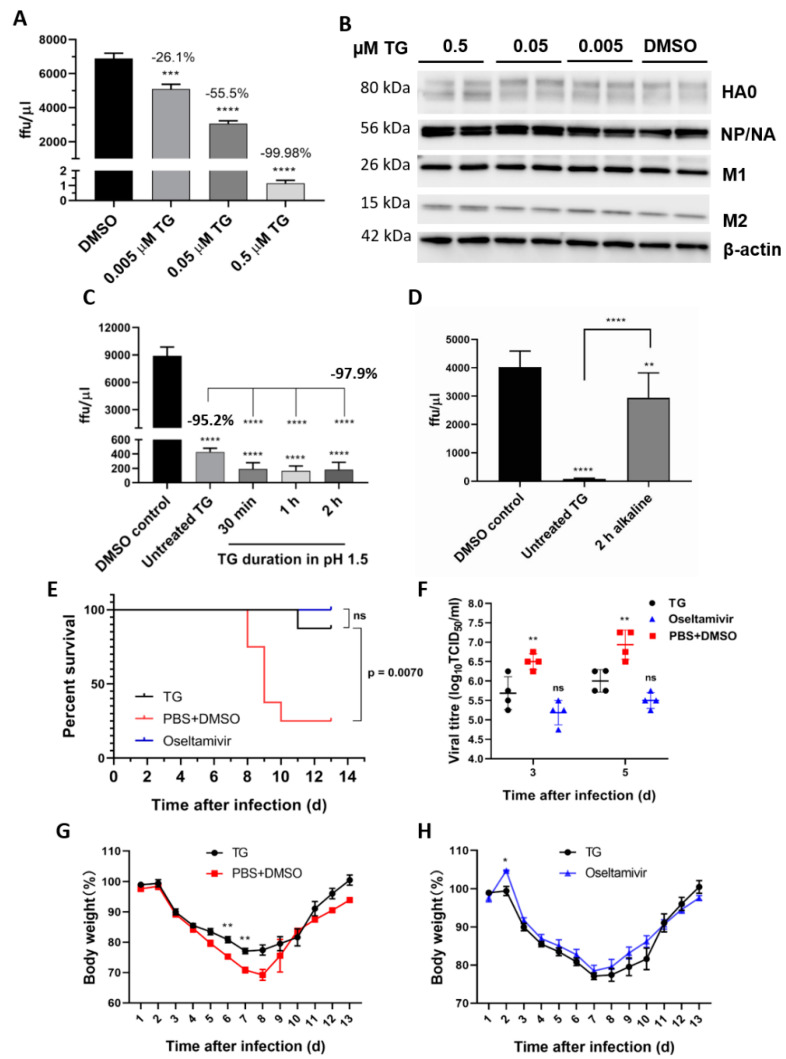
TG post-translationally inhibits the influenza A virus, is acid-stable and therapeutically protects mice in a lethal virus challenge. (**A**) A sharp reduction in influenza progeny production from the TG-primed NPTr cells was (**B**) accompanied by no detectable reduction in viral proteins. Cells were primed with TG for 30 min, washed twice with PBS and infected with USSR virus at 0.5 MOI for 2 h; after which cells were again washed 3 times with PBS and incubated in serum-free media, supplemented with TPCK trypsin at 0.2 µgl/mL, for 24 h. (**A**) A focus forming assay was performed with the corresponding media samples to determine viable virus production (ffu/µL). The indicated significance is based on one-way ANOVA (Dunnett’s multiple comparisons) and the percentage viable progeny relative to the corresponding DMSO control. (**B**) Viral proteins, including those processed through the ER–Golgi apparatus (HA, NA and M2), showed no reduction in TG-primed cells. (**C**,**D**) Antiviral activity of TG was stable in acidic but not alkaline condition. (**C**) TG used was first incubated in pH 1.5 (in 30 mM hydrochloric acid) for different durations, as indicated, and neutralised with sodium hydroxide before being applied to cells for 30 min at a 0.5 μM final concentration. (**D**) TG used was first incubated in pH 12.0 (10 mM sodium hydroxide) for 2 h and neutralised with hydrochloric acid before being applied to cells for 30 min at a 0.5 μM final concentration. Twenty-four hours post-infection with the USSR H1N1 virus at 0.5 MOI, the infected culture media were used in 6 h focus forming assays to immuno-detect the viral NP to determine the progeny virus output (ffu/μL). Unless otherwise indicated, the significance is based on one-way ANOVA Tukey’s multiple comparison and percentage virus reduction are in relation to the corresponding DMSO control. (**E**–**H**) Therapeutic protection of TG in a lethal influenza virus challenge in mice. Each BALB/c mouse (*n* = 8 per group) in a group was intranasally infected with 3 MLD_50_ of PR8/H1N1 virus. Each mouse was then dosed orally once a day with TG (1.5 μg/kg/day), oseltamivir (45 mg/kg/day) or PBS+DMSO for 5 days; the first dose was given at 12 hpi. Survival rates (**E**), viral titres in the lung by TCID_50_ assays (**F**) and body weight changes (**G**,**H**) were recorded over 14 d. Each time point represents the mean ± SEM. The Kaplan–Meier method was used for survival analysis. Significance indicated relative to the corresponding TG group. * *p* < 0.05, ** *p* < 0.01, *** *p* < 0.001 and **** *p* < 0.0001.

**Table 1 viruses-13-00234-t001:** Primer sequences.

Gene	Sense Primer (5′–3′)	Antisense Primer (5′–3′)
*18S* ribosomal RNA (universal)	ACGGCTACCACATCCAAGGA	CCAATTACAGGGCCTCG-AAA
*F* gene (RSV)	CAAGAACTGACAGAGGATGGTACTG	CATGTTTCAGCTTGTGGGAAGA
*L* gene (RSV)	AACACTTATCCTTCTTTGTTGGAACTTA	GCAACCGAAACTCACGATAGAAA
*M*-gene (RSV)	ACTCAAGAAGTGCAGTGCTAGCA	AAGGACACATTAGCGCATATGGT
*RIG-I* (human)	GAAGGCATTGACATTGCACAGT	TGGTTTGGATCATTTTGATGACA
*M*-gene (USSR H1N1)	AGATGAGCCTTCTAACCGAGGTCG	TGCAAAAACATCTTCAA-GTCTCTG
*M*-gene (pdm H1N1)	AGATGAGTCTTCTAACCGAGGTCG	TGCAAAGACACTTTCCA-GTCTCTG
*Orf1ab* (SARS-CoV-2)	CCGATCATCAGCACATCTAGGTT	GACAAGGCTCTCCATCT-TACCTTT
*Orf1ab* (OC43)	GCCAGGGACGTGTTGTATCC	TTGATCTTCGACATTGTGACCTATG

## Data Availability

The data presented in this study are available on request from the corresponding author.

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
