# Peer review of "Thapsigargin Is a Broad-Spectrum Inhibitor of Major Human Respiratory Viruses: Coronavirus, Respiratory Syncytial Virus and Influenza A Virus"

_viruses, 2021, doi:10.3390/v13020234_

Round 1

Reviewer 1 Report

This work in base on the previous work by the authors which indicates that hapsigargin (TG), an inhibitor of the sarcoplasmic / endoplasmic reticulum (ER) Ca2+ ATPase pump, induces a potent host innate immune antiviral response that blocks influenza A virus replication. In this manuscript, the authors expanded the antiviral effects of TG by testing it in RAV, coronavirus OC43, and SARS-CoV2. The results are confirmative and interesting, indicating TG is a broad-spectrum inhibitor of major human respiratory viruses.

Major:

  • One more schematic panel should be added in the figure to clarify the time window for the TG treatment relative to infection.
  • In most case the quantification in the figures are bars with SEM. In fig 1G,H and fig 7F, the quantification are shown with scattered blot indicating the distribution of the data. It would be nice to make all the quantification this way for displaying.
  • It is important to discuss more on the potential mechanism why TG, as an inhibitor of ER Ca2+ ATPase pump, could serve as more efficient antiviral effects, in addition to the ER stress and UPR.
  • There are totally 7 human infected coronaviruses, the authors tested CoV-2 and OC43. How about others, such as 229E, NL63, if the author try to claim it is broad-spectrum inhibitor of major human respiratory viruses.
  •  

Minor:

  • The annotation in Fig 1B and 1D is not consistent, ‘0.1uM TG’ vs ‘TG 0.1uM’.
  • In fig 4, the size of the label is different and small, some are bold while some are not.

Reviewer 2 Report

Al-Beltagi et al. conducted an in vitro study on the impact of Thapsigargin (TG) on coronavirus, RSV and influenza A virus, and concluded that TG inhibited the above-mentioned viral infection with therapeutic potential and was superior to current medications, e.g. Ribavirin.

In general, I found this topic important and the careful thoughts by the authors, which was well presented in the manuscript. However, I’ve certain concerns on the study design and reported data, especially with regards to cell viability of treatment groups and the regime of TG priming, which makes the conclusion less convincing as expected.

I would therefore recommend publication with major revision, based on the reasons detailed as follows.

  • Major Comments
  1. My major concern is with regards to cell viability of the treatment groups, which was not adequately presented. As viral replication would naturally cease in dead cells, and there is known cytotoxicity of TG to cells, it’d be necessary to present that TG did not affect cell viability in all treatment regimes, not only figures 1G and 1H.

  1. Figure 1, the authors kept on changing the regime of priming cells with TG. For example, Panel E, why did the author choose to TG prime the cells immediately prior to infection, which was not the same as Panels A and B? Panel F, why 48 hours pre-infection which was also a new regime? What happened if you follow the timing of TG priming as Panels A and B? Similarly, the authors presented multiple doses of TG on A549, but only 2 doses on HEp2. What happened to the rest of the doses (0.01 uM, 0.025uM and 0.05uM)? If these were not tested, why?

Round 2

Reviewer 1 Report

The answers are satified. 

Reviewer 2 Report

This is an important topic, and I would like to thank the authors for carefully addressing my concerns from the previous version of manuscript, especially with regards to the cell viability and treatment regimes. I therefore recommend this paper for publication without further revision.